# The Complex Histopathological and Immunohistochemical Spectrum of Neuroendocrine Tumors—An Overview of the Latest Classifications

**DOI:** 10.3390/ijms24021418

**Published:** 2023-01-11

**Authors:** Ancuța-Augustina Gheorghișan-Gălățeanu, Andreea Ilieșiu, Ioana Maria Lambrescu, Dana Antonia Țăpoi

**Affiliations:** 1Department of Cellular and Molecular Biology and Histology, Carol Davila University of Medicine and Pharmacy, 050474 Bucharest, Romania; 2C.I. Parhon National Institute of Endocrinology, 011863 Bucharest, Romania; 3Department of Pathology, Carol Davila University of Medicine and Pharmacy, 050474 Bucharest, Romania; 4Department of Pathology, University Emergency Hospital, 050098 Bucharest, Romania; 5Victor Babes National Institute of Pathology, 050096 Bucharest, Romania

**Keywords:** neuroendocrine tumors, neuroendocrine neoplasms, carcinomas, histopathological classification, tumor grade, Ki67

## Abstract

Neuroendocrine neoplasms (NENs) originate from the neuroendocrine cell system, which may either take the shape of organoid cell aggregations or be composed of dispersed cells across various organs. Therefore, these tumors are heterogenous regarding the site of origin, functional status, degree of aggressiveness, and prognosis. When treating patients with neuroendocrine tumors, one of the most significant challenges for physicians is determining the correct tumor grade and thus classifying patients into risk categories. Over the years, the classification of these tumors has changed significantly, often causing confusion due to clinical, molecular, and immunohistochemical variability. This review aims to outline the latest NENs classifications regardless of their site of origin. Thus, an overview of the key histopathological and immunohistochemical characteristics of NENs could pave the way to validate possible predictive and prognostic markers and also guide the therapeutic conduct.

## 1. Introduction

Neuroendocrine tumors originate from neuroendocrine cells of the diffuse endocrine system. These cells produce peptide hormones stored in neurosecretory granules and released into the bloodstream [1,2]. Neuroendocrine cells share some common ultrastructural (membrane bound secretory vesicles of varying electron density, abundant rough endoplasmic reticulum and free ribosomes and electron-dense mitochondria) [2,3] and cytochemical characteristics (silver reactivity). Following his discovery of thyroid parafollicular cells, these characteristics prompted A.G.E. Pearse to develop the APUD (Amine Precursor Uptake and Decarboxylation) cell theory and coin the term APUDoma for tumors of APUD cells throughout the body [4,5]. At that time, A.G.E. Pearse assumed that all APUD cells originate from neural crests [4]. However, it is now established that all cells, which share this common structure and function, comprise the diffuse neuroendocrine system. Depending on their embryological origin, these cells may be divided into non-epithelial neuroendocrine cells or epithelial neuroendocrine cells (derived from the neuroectoderm). Epithelial neuroendocrine cells are the most widespread and can be located in both endocrine organs such as the anterior pituitary; the parathyroid (chief cells); and the thyroid (C cells), and diffuse neuroendocrine tissue of the lung, breast, gastrointestinal system, pancreas, genitourinary system, and skin (Merkel cells) [2]. Non-epithelial neuroendocrine cells represent paraganglion cells associated with either sympathetic or parasympathetic nervous systems in various body parts [6]. Sympathetic paraganglion cells are mainly located in the abdomen, in the connective tissue along the vertebral column, the adrenal medulla, urogenital system, organ of Zuckerkandl [6], gallbladder, and liver [7]. Parasympathetic paraganglion cells are located in the head and neck, in association with cranial nerves and their thoracic branches of the glossopharyngeal and vagus nerve, the carotid body, in the tympanic glia (also known as “glomus jugulare”), the laryngeal paraganglia, the paraganglia of the lungs, and heart [6]. Regardless of their localization and embryological origin, all the neuroendocrine cells can be the source of neuroendocrine tumors, representing an extraordinarily diverse group of neoplasms with variable clinical manifestations, pathological features, and prognosis (Figure 1).

Due to the heterogeneity of these neoplasms, several different terms have been used to define them. For example, epithelial neuroendocrine neoplasms include tumors described as carcinoid, neuroendocrine tumors, neuroendocrine carcinoma, small-cell carcinoma, thyroid medullary carcinoma, islet cell tumor, Merkel cell carcinoma, argentaffinoma, or APUDoma. In contrast, non-epithelial neuroendocrine neoplasms include paraganglioma, pheochromocytoma, and neuroblastoma [8]. Nevertheless, they all share common immunohistochemical expression of several markers such as chromogranins, synaptophysin, CD56, and Insulinoma-associated protein 1 (INSM1) [8,9,10]. Therefore, it is now accepted that all tumors displaying predominant neuroendocrine differentiation, regardless of their anatomic site or other histologic features such as tumor grade, should be referred to generally as neuroendocrine neoplasms (NENs) [8]. In addition to this, based on histologic criteria, NENs can be further divided into well-differentiated neoplasms called “neuroendocrine tumors” (NETs) and poorly differentiated neoplasms called “neuroendocrine carcinomas” (NECs) [8,11]. Therefore, the primary objectives of this review are to comprehend the key histopathological characteristics of NENs, with emphasis on the grading system, and to define the most prevalent organ-specific NEN.

## 2. Histopathological and Immunohistochemical Features in NENs

As previously stated, NENs are classified according to tumor grade into well-differentiated neoplasms (NETs) and poorly differentiated neoplasms (NECs). This dichotomous classification is consistent with significant genetic differences, risk factors, clinical features, and prognosis [8]. NETs often exhibit an “organoid” cell proliferation with sporadic nests, trabecular patterns, and glandular or rosette development [1,11]. Tumor stroma is usually scant with prominent blood vessels. Occasionally, cell palisading near the periphery of the nests, amyloid stroma, and calcifications with psammoma bodies are seen. The tumor cells are usually epithelioid and closed-packed, but spindle and dis-cohesive cells have also been reported. The cells are medium-sized, with abundant cytoplasm, and uniform round to oval nuclei with inconspicuous nucleoli. The nuclei have characteristic fine to coarsely granular chromatin, giving the classic “salt and pepper” aspect. Mitotic figures are absent or rare [11]. Epithelial NETs express keratins such as AE1/AE3 or CAM5.2. In addition, these cells also display intense immunohistochemical positivity for neuroendocrine markers such as synaptophysin, chromogranin A, somatostatin receptors (SSTRs), or CD56 [11,12,13].

NECs, on the other hand, present as a solid proliferation of less monomorphic cells with either scant (small-cell types) or abundant (large-cell types) cytoplasm, irregular nuclei with severe nuclear molding, and high mitotic rates [11,12]. Small-cell NECs display hyperchromatic nuclei with “salt and pepper” chromatin, while large-cell NECs exhibit vesicular nuclei with conspicuous nucleolus, which can be large and intensely eosinophilic [11]. Areas of necrosis and apoptotic bodies are commonly reported in NECs [1,11]. In addition, NENs, and especially NECs, can also exhibit a non-neuroendocrine component with various differentiation, such as glandular or squamous [8]. The significant histopathological differences between NETs and NECs are summarized in Table 1.

Immunohistochemical positivity for cytokeratins and at least one neuroendocrine marker are required to confirm the diagnosis of NEN. Two distinct neuroendocrine markers are necessary to validate the diagnosis of large-cell NECs [11]. Immunohistochemistry for synaptophysin, SSTRs, and chromogranin A is less useful since their expression is weaker in NECs [11,12,13]. Most often, NECs remain positive for CD56, but this marker should be interpreted with caution because of its low specificity [13]. As a result, “second-generation” neuroendocrine indicators are now being investigated. INSM1 has emerged as a crucial novel marker for neuroendocrine differentiation, being more sensitive than synaptophysin or chromogranin A and being expressed in both NETs and NECs in many organs. Additionally, its nuclear expression, as opposed to the other neuroendocrine markers with cytoplasmatic expression, makes it easier to interpret [10,14,15,16,17,18]. However, INSM1 should not be used as a stand-alone marker for neuroendocrine differentiation, since its expression can be positive in other types of tumors, such as lung squamous cell carcinomas, lung adenocarcinomas, lymphomas, and several different types of sarcomas [19,20,21,22]. Another related marker, insulin gene enhancer protein Islet-1 (ISL1), is expressed in various NENs located in the pancreas, duodenum, colorectum, skin, thyroid, and also in pheochromocytomas/paragangliomas and retains its sensitivity even in neuroendocrine carcinomas [23,24]. The major limitation of ISL1 is its lack of expression in neuroendocrine neoplasms of the small intestine and the appendix [24]. However, combined with other neuroendocrine markers, it may be used to exclude the small intestine as the primary site of a metastatic NEC [25]. The great sensitivity of secretogranin (SECG) expression in NETs and NECs in several organs, including colorectal NENs where chromogranin A is typically negative, has also been documented. Nevertheless, using SECG also has limitations, as this marker is not usually expressed in pheochromocytomas and paragangliomas [23,24]. CD200, a marker for hematopoietic neoplasms, has recently become of interest for neuroendocrine neoplasms. While research is still limited in this field, several authors have demonstrated that CD200 could be a sensitive immunohistochemical marker for neuroendocrine neoplasms of various sites. Additionally, it has been shown that there is a correlation between C200 expression and tumor grade; as a result, CD200 should be considered a potential therapeutic target [15,26].

## 3. Histologic Grading of NENs

The classification of NENs according to tumor grade is of the utmost importance since this grading is correlated with the prognosis and therefore influences the therapeutic approach. However, this classification is often the source of great confusion for clinicians, as the defining criteria and terminology of each entity are not universal for every organ and have changed significantly throughout the years [12]. At the beginning of the 20th century, Oberndorfer S. introduced the term “carcinoid”, describing a series of small intestine benign tumors composed of argentaffin-positive and argyrophilic cells [27]. Later in the 20th century, carcinoid tumors were reported in other organs, and some were proven to have aggressive behavior [28,29]. Finally, in 1980, the World Health Organization (WHO) described all NETs, except tumors of endocrine glands and pulmonary neuroendocrine tumors, as carcinoids. This classification caused a lot of controversies, since pathologists referred to all diffuse neuroendocrine tumors as “carcinoids”, although clinicians only associated it with patients displaying symptoms of carcinoid syndrome [12,30]. In 1999, the WHO introduced the terms typical carcinoid, atypical carcinoid, and NEC (either small-cell or large-cell) for pulmonary NETs [12]. However, as the term “carcinoid” remains confusing, the WHO intends to replace it altogether, and its first attempt to do so was the 2000 classification for gastroenteropancreatic NENs. This classification defined the following entities: well-differentiated NETs with presumably benign behavior, well-differentiated NETs with uncertain behavior, and poorly differentiated NECs with high-grade malignant behavior [31]. Since then, the WHO has issued several additional classifications for NENs, although inconsistencies in reporting these lesions have continued due to variances in the criteria for each organ system [8]. Therefore, in 2018, the WHO proposed a standard classification for NENs, regardless of their site of origin [8].

First and foremost, the expert group defined all tumors with predominant neuroendocrine differentiation based on immunohistochemical criteria, whether well or poorly differentiated, as NENs. Furthermore, based on histological criteria, epithelial NENs should be classified as well-differentiated and poorly differentiated. Overall, NETs exhibit non-neoplastic histology to a large extent, while NECs have high-grade histological characteristics. Thus, “neuroendocrine tumor” and “neuroendocrine carcinoma” should be used for well-differentiated NENs and poorly differentiated NENs, respectively. In addition, specific tumor characteristics must be reported for selected organs (e.g., carcinoid tumor, small-cell pulmonary NEC and large-cell pulmonary NEC for lung neuroendocrine neoplasms).

Finally, a grading system was recommended to be implemented for most NENs. According to this system, NETs should generally be graded as G1, G2 and G3, representing low-grade, intermediate-grade and high-grade, respectively. NECs need no further grading, as they are, by definition, high-grade [8]. To grade NENs, three parameters must be reported: mitotic count, Ki67 labeling index, and the presence or absence of necrosis. However, tumor necrosis is not a grading criterion for gastrointestinal or pancreaticobiliary tumors. The mitotic count should be expressed as mitoses per mm^2^ in up to 10 mm^2^ rather than as mitoses per high-power field (HPF) due to variations in HPF area in microscopes. The Ki67 index must be reported in hotspots of intense labeling of at least 0.4 mm^2^ [8,11]. In 2022, the WHO published its latest Classification of Endocrine and Neuroendocrine Tumors, officially supporting the aforementioned grading system for most NETs; nevertheless, there are still differences in reports for each system. For example, NETs of the gastrointestinal and pancreaticobiliary tract, as well as NETs of the upper aerodigestive tract and salivary glands, are graded using a three-tiered system. However, the grading criteria can differ [8]. In contrast, lung and thymus NETs are classified as typical low-grade carcinoids and intermediate-grade atypical carcinoids corresponding to G1 and G2 grades, respectively. Nevertheless, some authors reported cases of atypical lung carcinoids with high mitotic counts (>10 mitoses/2 mm^2^ and Ki67 > 30%), which could be regarded as an equivalent to NET G3 of the gastrointestinal tract [32,33]. NECs are not subclassified further based on the mitotic activity in any of these organs. Thyroid medullary carcinoma, which is further classified as low-grade or high-grade, is an exception. The major defining criteria for all these entities are presented in Table 2, according to the 2022 WHO Classification of Endocrine and Neuroendocrine Tumors [34].

## 4. Differences between NET G3 and NEC

Recognizing NETs G1 and G2 is usually straightforward, as these tumors display well-differentiated morphological features and low-to-moderate mitotic counts. Distinguishing between these two entities is based on the proliferative rate. The most problematic situations arise from the difficulty distinguishing NETs G3 from NEC, in which proliferative rates alone do not represent the diagnostic criteria [35]. For example, Ki67 is not a marker that can distinguish between the two, yet NET G3s have lower Ki67 (mean values around 40%) than NEC G3s (mean values > 70%) [36]. Therefore, the definitive diagnosis relies mainly on the morphological features, which can sometimes be challenging to tell apart, particularly on small biopsies. In such cases, NET G3s could be misdiagnosed as NECs due to significant cellular atypia and areas of necrosis [37,38]. To establish the correct diagnosis, in addition to pathology examination, clinical and imaging features should also be considered. For example, patients with NECs present with nonspecific symptoms of an aggressive systemic illness and have undetectable chromogranin levels, rarely exhibiting hormonal symptoms [11,37]. In 2015, Heetfeld and colleagues published a multicentric retrospective study of 204 patients with NETs G3 and NEC. The study’s results revealed that patients with NET G3 were more likely to have a functional tumor (*p* = 0.003) than those diagnosed with NEC [39]. In terms of biomarkers such as chromogranin A, urinary 5HIAA, and neuron-specific enolase (NSE), no statistically significant difference was observed between NET G3 and NEC [40].

Epidemiological data are scarce regarding NET G3. However, in contrast to NEC, the primary location for NET G3 is the pancreas, with better overall survival [40,41]. Thus, NET G3 is closer to well-differentiated NET G2 than NEC in terms of prognosis and response rate but with worse overall survival [42].

In imaging studies, NETs can be detected on functional imaging, as they can remain undetected on PET/MRI. Therefore, fluorodeoxyglucose-PET (FDG-PET) is recommended in NEC as part of functional imaging [42]. The clinical and imaging differences can be correlated with some immunohistochemical features of these two entities. NETs are typically positive for specific hormones and strongly express SSTRs [43,44]. In contrast, NECs do not express specific hormones and, except for a few large-cell NECs, are mostly negative or weakly positive for SSTRs [8,43,45]. A positive FDG-PET does not appear to be able to differentiate NEC from NET G3. However, the latter is more likely to have positive somatostatin receptor imaging [40].

Finally, another characteristic that proves the two entities do not overlap is the disparity in response to platinum–etoposide regimens. Thus, Velayoudom-Cephise et al. reported that 31% of large-cell NEC patients had an objective response to platinum–etoposide versus 0% for the NET G3 group [40].

## 5. Pathological Features of Mixed Neuroendocrine–Non-Neuroendocrine Neoplasms

In addition to pure NENs, pathologists can encounter tumors composed of neuroendocrine and non-neuroendocrine elements. These neoplasms are defined as mixed neuroendocrine–non-neuroendocrine neoplasms (MiNENs) and must distinctively include both components, each with its characteristic morphology and immunohistochemical pattern [11]. These distinctive features must be noted to distinguish MiNENs from other malignancies with scattered neuroendocrine cells or aberrant neuroendocrine expression [11]. Moreover, it is not yet clear what is the exact proportion of each component to make the positive diagnosis of a MiNEN. For gastrointestinal neoplasms, a neuroendocrine component cutoff value of 30% is required for a MiNEN diagnosis. However, the validity of this value is questionable, since it has been proven that a high-grade component has a significant prognostic value even when present in a lower percentage [46]. For other organs, the 2022 WHO Classification of Endocrine and Neuroendocrine Tumors does not list the 30% cutoff criteria for diagnosing MiNENs. Therefore, this value remains in use solely for diagnosing digestive MiNENs, but further studies are needed to determine its predictive value [11]. In light microscopy, the neuroendocrine component of MiNENs is usually represented by NEC, either small or large cells, each exhibiting the characteristic features already mentioned above. Less frequently, the neuroendocrine component can show features of a well-differentiated NET [11]. Along with the distinctive morphology of a MiNEN, the neuroendocrine component must be confirmed by immunohistochemistry expression of specific neuroendocrine markers. Moreover, the Ki67 proliferation index should also be assessed to grade the neuroendocrine component in cases of mixed tumors [11]. In cases when the neuroendocrine component is a NEC, the Ki67 proliferation index was shown to be a prognostic factor for lung and digestive MiNENs [47,48]. The non-neuroendocrine component is generally represented by the most common subtypes of carcinomas usually found in the affected organ (adenocarcinoma or squamous cell carcinoma) [11].

## 6. Characteristics of Epithelial NENs Based on the Organ of Origin

### 6.1. Lung NENs

For lung NENs, additional grading features and the distinction between well-differentiated NETs and poorly differentiated NECs need further clarification. In 2021, the WHO Classification of Thoracic Tumors classified lung NENs as typical carcinoid, atypical carcinoid, and high-grade carcinoma with small or large cells [49]. According to this classification, distinguishing between all these entities relies on cytological features, mitotic count, and whether tumor necrosis is present. The cutoff value of 10 mitoses/2 mm^2^ is the criterion that separates carcinoids from carcinomas [49]. Though not mandatory, the latest WHO classification considers Ki67 a “desirable” feature to be reported, particularly on small biopsies for differentiating carcinoids from carcinomas [49,50]. In this respect, on biopsy samples, Ki67 better predicts the definitive proliferative activity of the resected tumor than the mitotic count [51]. The cutoff value of the Ki67 index used to define each category of lung NENs, and its prognostic value are still up for debate [8]. Some authors recommend Ki67 index assessment in lung NETs [52,53], while others state that it has no particular predictive value [54]. A novel concept of tumors with well-differentiated morphology but high mitotic counts and Ki67 proliferation rates, comparable to G3 NETs of the digestive tract, has evolved as a result of all the recent attempts to establish a classification for lung NENs that may be useful for diagnosis and prognosis [55,56]. However, this category is not officially recognized by the WHO Classification. Therefore, such tumors are classified as NECs [8]. Similar to NECs, these tumors exhibit over 10 mitoses/2 mm^2^ and a Ki67 index > 20% in hotspot areas [55]. Nevertheless, molecular studies suggest that these tumors are related to carcinoids rather than NECs as they present characteristic mutations for carcinoids (MEN1) and lack distinct mutations for NECs (Rb1 and TP53) [32,33,57]. From a clinician’s point of view, it is essential to acknowledge the existence of well-differentiated lung NETs with high proliferation rates because of their distinctive behavior. Although they resemble the clinical course of carcinoids rather than NECs, they tend to be more aggressive and have high recurrence rates [56,58]. Therefore, it is likely that the Ki67 index will become a mandatory feature to be assessed to grade lung NETs, similar to those localized in the digestive tract [32,53].

Small-cell lung carcinomas (SCLCs) are usually easily diagnosed on the standard histopathological exam due to their classic features: small cells with scant cytoplasm with indistinct borders and characteristic nuclei and high mitotic counts of more than 10 mitoses/2 mm^2^. In addition, extensive necrosis and the Azzopardi effect are distinctive [49]. Due to these features, the diagnosis of SCLCs can be frequently established without immunohistochemistry. Nonetheless, immunohistochemical tests may increase diagnostic accuracy [59]. Since around 15–20% of SCLCs are negative for synaptophysin and chromogranin A, additional markers may be used [13]. In addition, most SCLCs express CD56 and/or INSM1 [18,60,61]. In rare cases when all these markers are negative [18] but the diagnosis of SCLC is still morphologically favored, the final diagnosis is established by excluding mimickers with the use of other markers such as CD45 for lymphoma or p40 for the basaloid version of squamous cell carcinoma [13]. In addition, a novel marker, POU2F3, is present in SCLCs that lack expression of existing neuroendocrine markers [62,63].

Large-cell neuroendocrine carcinomas (LCNECs) lack the traditional features of NECs. Instead, they comprise large cells with abundant cytoplasm and vesicular nuclei with prominent nucleoli. However, LCNECs usually display classic neuroendocrine architecture, such as nests with palisading, trabeculae, and rosettes [49]. In this context, the main differential diagnosis is poorly differentiated non-small-cell lung carcinomas, which makes confirming the neuroendocrine origin using immunohistochemistry mandatory [13].

### 6.2. Gastrointestinal and Pancreatobiliary NENs

The 2019 WHO Classification of Digestive System Tumors classifies NENs as well-differentiated NETs and poorly differentiated NECs based on morphological criteria [64]. Furthermore, NETs are subclassified based on mitotic count and Ki67 index into low-grade NET G1, intermediate-grade NET G2, and high-grade NET G3. In contrast, NECs are subclassified as either small-cell NEC or large-cell NEC [64]. The ultimate diagnosis for NETs with disparities between the mitotic count and Ki67 index should correspond to the higher grade [1]. According to the WHO classification, cytological characteristics—rather than proliferative activity—are used to distinguish G3 NETs from NECs [64]. Nevertheless, NECs usually show significantly higher proliferation rates than G3 NETs, and the Ki67 assessment can aid in establishing the diagnosis [65,66]. Another difference between these two entities is the expression of SSTRs, which is diffusely positive in NETs but mostly absent in NECs [67]. In addition, immunohistochemistry for p53 and RB1 may be helpful as NECs display either aberrant p53 expression (hyperexpression or null) or null RB1 expression [68]. In contrast, NETs display a wild-type p53 and diffuse RB1 expression [37]. Ultimately, in NETs, chromogranin A expression is diffusely and intensely positive, while in NECs, it is focally and weakly positive [38].

In addition to these general characteristics of gastrointestinal and pancreatobiliary NENs, there are also some site-specific features. For gastric NENs, there are three classic subtypes, as described by Rindi et al. [69], with a fourth subtype observed more recently [70]. Type 1 NENs are the most common. They are associated with chronic atrophic gastritis and hypergastrinemia. The origin of these tumors is represented by enterochromaffin-like (ECL) cells [1,70]. These tumors are generally multifocal, but the prognosis is favorable, with very few cases of metastasis reported [71]. Type 2 NENs occur in patients with Zollinger-Ellison syndrome and multiple endocrine neoplasia (MEN). They also originate from ECL cells [70] and are usually G1 or G2 NETs [1], with an overall good prognosis [72]. Type 3 NENs occur sporadically, are not associated with other conditions, and probably originate from ECL cells [70]. This neoplasm is usually solitary and large, with high metastasis rates [1,70,73]. Lastly, type 4 NENs are the rarest, and their cell of origin remains unknown [70]. They are large lesions with aggressive behavior and poor prognosis [74].

NENs have different characteristics throughout the small intestine depending on the affected region. Duodenal NENs can also be classified as well well-differentiated duodenal NETs (Duo-NETs), poorly differentiated neuroendocrine carcinomas (NECs), mixed neuroendocrine–non-neuroendocrine neoplasms (MiNENs), and gangliocytic paragangliomas [75]. Duodenal NETs are mostly G1, but G2 and G3 have also been described [75]. Based on clinical and pathological features, three types of epithelial duodenal NETs have been described: ampullary-type somatostatin-producing NETs (AS-NETs), gastrinomas, and ordinary nonfunctioning NETs [76]. AS-NETs display a tubulo-acinar/pseudoglandular growth pattern, which can cause them to be misdiagnosed as adenocarcinoma. However, AS-NETs are characterized by psammoma bodies’ presence, neuroendocrine markers’ expression, and SSTRs [75]. The most frequently diagnosed small bowel NENs are represented by well-differentiated serotonin-producing enterochromaffin cell neuroendocrine tumors (EC cell-NETs) located in the distal ileum, followed by gastrin-producing NETs, primarily found in the jejunum. Jejunal-ileal NENs are usually low grade, G3 NETs being rarely reported. NECs are MiNENs are also rarely located in these sites [75].

Appendiceal NETs are the most frequent neoplasms of the appendix, with an incidental diagnosis during appendectomy [77]. They are usually of low or intermediate grade [78] and have an excellent prognosis [64]. Pure NECs are extremely rare and resemble colonic NECs, either the small-cell or large-cell subtype [60]. A more aggressive course characterizes colorectal NENs compared to other digestive NENs [12]. Their prognosis largely depends on tumor stage and grade [63], but rectal NETs, especially G3 (larger than 1 cm), have a poor prognosis [1,64]. Anal NENs are rare entities [77], usually G1 or G2 [79]. Localized NETs have an excellent prognosis [64,77]. The diagnosis of anal NEC can sometimes be challenging, as it can be misdiagnosed with skin cancers such as basaloid squamous cell carcinoma, basal cell carcinoma, melanoma, or Merkel cell carcinoma [77]. Merkel cell carcinoma, in particular, can be difficult to differentiate as it also expresses neuroendocrine markers. However, Merkel cell carcinoma cells stain positive for CK20 with a distinctive dot-like pattern [80].

Pancreatic NENs are classified into well-differentiated NETs and poorly differentiated NECs. They are all malignant except for the pancreatic neuroendocrine microadenoma, a nonfunctioning NET smaller than 0.5 cm [1,64]. In addition, NETs are divided into functioning and nonfunctioning tumors based on hormone secretion and clinical symptoms. Functioning NETs are most commonly insulinomas, followed by gastrinomas [12]. Other functioning NETs include VIPomas, glucagonomas, and rare types such as somatostatinomas, ACTH-producing NETs, and serotonin-producing NETs [64,77]. Insulinomas have the best prognosis, but the others are aggressive and often metastasize [1,64]. Another challenge in diagnosing pancreatic NENs is differentiating pancreatic NECs or G3 NETs from an acinar cell carcinoma. In such cases, immunohistochemistry for BCL10 and trypsin is mandatory for the final diagnosis [38,64].

### 6.3. Head and Neck NENs

The 2016 WHO Classification of Head and Neck Tumors uses different terms for defining NENs depending on their localization. The only recognized entity for the nasal cavity, paranasal sinuses, and skull base is neuroendocrine carcinoma, a poorly differentiated carcinoma with either small- or large-cell morphology [81]. The differential diagnosis of this rare entity includes a variety of other tumors, such as undifferentiated sinonasal carcinoma, NUT carcinoma, mucosal melanoma, embryonal rhabdomyosarcoma, neuroblastoma, or paraganglioma. Thus, a thorough immunohistochemical analysis should be performed to establish the correct diagnosis [8].

For the hypopharynx, larynx, trachea, and parapharyngeal space, the WHO defines the following entities: well-differentiated NEC (also referred to as typical carcinoid or neuroendocrine carcinoma G1), moderately differentiated NEC (also referred to as atypical carcinoid, neuroendocrine carcinoma G2) and poorly differentiated NEC, small or large-cell [81]. The distinction between well-differentiated and moderately differentiated carcinomas is made based on assessing the presence of necrosis and mitotic counts. Necrosis is absent in well-differentiated NEC but present in moderately differentiated cases. The cutoff for mitoses is 2 mitoses/2 mm^2^ [81]. The Ki67 index is not an established criterion for grading these tumors [8].

Chromogranin A and synaptophysin may display weak immunohistochemical expression, particularly in NECs of the head and neck. Therefore, INSM1 may be particularly useful for establishing the diagnosis in such cases [81].

Thyroid NENs are represented by medullary thyroid carcinomas (MTC) [82]. The 2022 WHO Classification of Endocrine Tumors introduced a grading system based on the mitotic count, Ki67 index, and the presence of necrosis. Thus, low-grade MTC display <5 mitoses/2 mm^2^, Ki67 < 5%, and no necrosis. High-grade MTC must display at least one of the following features: ≥5 mitoses/2 mm^2^, Ki67 ≥ 5% or necrosis [83]. In addition to these features, weak immunohistochemical expression of calcitonin may be associated with a less favorable prognosis [8].

### 6.4. Breast NENs

Breast NENs are rare neoplasms segregated by the WHO Classification of Breast Tumors into NETs G1, NETs G2, and NECs, small and large cells [84]. Breast NETs can be particularly challenging to diagnose since other breast carcinomas can express neuroendocrine markers. Therefore, the distinction between these entities should primarily be based on histological criteria [8,84]. Presently, further grading of NETs as G1 or G2 does not have well-established criteria, and the Nottingham score can be used according to the WHO classification [84]. In addition, Ki67 rates and necrosis assessment are not clearly defined for evaluating these tumors [8].

### 6.5. Femele Reproductive System NENs

NENs are rare neoplasms of the female genital tract. The most affected site is the cervix, followed by the ovary [12]. According to the 2020 WHO Classification of Female Genital Tumors, they are classified as NETs, G1 or G2, and NECs, small- or large-cell types, in all sites except for the ovary, where the term “carcinoid tumor” is still used [85]. This distinction exists because ovarian NENs are well-differentiated and have an excellent prognosis. NETs can occur in any part of the female genital system. However, in the endometrium and cervix, NECs are much more prevalent than NETs [85]. To establish the diagnosis of NENs in the female reproductive system, immunohistochemical positivity for at least one neuroendocrine marker (chromogranin A, synaptophysin, or CD56) is required. In cases expressing only CD56, the diagnosis of NEN should only be considered in the presence of clear histopathological neuroendocrine features [85].

### 6.6. Urinary and Male Genital System NENs

Like in most systems, the 2022 WHO Classification of Urinary and Male genital Tumors classifies NENs as NETs, G1 or G2, and NECs, small- or large-cell types [86]. Like other NENs, these tumors express neuroendocrine markers (chromogranin A, synaptophysin, or CD56). Nevertheless, SCNEC lacking chromogranin A or synaptophysin can still be diagnosed based on morphology alone [86]. INSM1 may also be used as it has superior sensitivity to other neuroendocrine markers [86]. The most common lesions in these organs are SCNECs, followed by LCNECs. NETs are extraordinarily rare, and no definitive grading criteria or prognostic factors have been established [87]. In the testis, they may originate from a teratoma or other germ stem tumors [85]. SCNECs often cause a de novo occurrence in the prostate and the bladder [88]. Prostate SCNECs usually represent a transdifferentiation phenomenon in prostate carcinomas treated with anti-androgenic hormones [89]. In such patients, transdifferentiation can also lead to a prostatic LCNEC [87].

### 6.7. Skin NENs

Skin NENs are high-grade carcinomas, also called Merkel cell carcinomas (MCC). MCC can display small- and large-cell features, but the most common type is the intermediate cell. Consequently, the WHO does not officially subclassify MCC [8,90]. MCCs express neuroendocrine markers (chromogranin A, synaptophysin, or CD56) and display characteristic perinuclear dot-like staining for CK20 and neurofilament (NFT) [90]. Furthermore, the Ki67 index, and necrosis are not used as prognostic factors [8]. Instead, tumor size is the most important prognostic factor, and tumor-infiltrating lymphocytes may also be a prognostic factor [90].

Well-differentiated NENs of the skin are most likely metastatic lesions [8]. However, cases of primary skin well-differentiated NENs have been reported [91] but they should rather be regarded as low-grade sweat gland carcinomas, sebaceous neoplasms, or basal cell carcinomas with neuroendocrine differentiation [8].

The major immunohistochemical features of NENs discussed so far in chapter 6 are summarized in Table 3.

### 6.8. Pituitary NENs

Adenohypophysial hormone-secreting tumors have historically been referred to as “pituitary adenomas”. Although these tumors can spread locally and even metastatically, no morphologic features predict their aggressiveness [2,8]. In addition, the tumor cells of pituitary adenomas express neuroendocrine markers such as CD56, synaptophysin, chromogranin A, and INSM1. Consequently, the latest WHO Classification of Endocrine and Neuroendocrine Tumors now defines them as pituitary neuroendocrine tumors (PitNETs) and, therefore, are considered malignant [83]. The WHO does not recommend the term “pituitary carcinoma” since no definitive clinical or morphological criteria can accurately predict metastatic potential [2,92]. Therefore, the WHO favors using the term metastatic PitNETs in the presence of spread beyond the point of origin [83].

In the 2022 edition of WHO Classification of Endocrine and Neuroendocrine Tumors, PitNETs are classified based on the immunohistochemical expression of three transcription factors: (TPIT, PIT1, and SF1), involved in the differentiation of the cell types that give rise to various tumors [83]. Thus, it is recommended to classify PitNETs based on cell lineage according to transcription factors rather than on hormone immunohistochemistry, as this can sometimes be nonspecific [93]. In this respect, diagnosing multiple synchronous PitNETs of different lineages is only possible by analyzing the expression of transcription factors [94]. Therefore, in addition to the transcription factors mentioned above, the definitive classification of PitNETs requires the analysis of other transcription factors (GATA3 and ER-alpha); hormones (GH, PRL, ACTH, beta-TSH, beta-LH, beta-FSH, and alpha-subunit of glycoprotein hormones) and low molecular weight keratins (CAM5.2 and CK8/18) [83,95]. Based on these criteria, the 2022 WHO classification of PitNETs is presented in Table 4.

Ki67 proliferation index is not mandatory, as its predictive value for diagnosing aggressive tumors is still under debate [96]. The prognosis of PitNETs depends on tumor size, type, and local invasion [84,97,98]. Particularly aggressive tumors include immature PIT1 lineage [83], Crooke cell [99], “silent” corticotroph [83], sparsely granulated corticotroph [100,101], and null cell tumors [102,103].

## 7. Non-Epithelial Neuroendocrine Neoplasms

Non-epithelial NENs are derived from neural crests and are called phaeochromocytomas when located in the adrenal medulla and paragangliomas when they arise from the extra-adrenal autonomic paraganglia. Pheochromocytomas may also be referred to as intra-adrenal sympathetic paragangliomas. The WHO Classification of Endocrine and Neuroendocrine Tumors, published in 2022, states that non-epithelial NENs are all malignant by definition. However, their prognosis is highly variable and is based on several factors currently under debate [83].

Classic phaeochromocytoma/paraganglioma histology is characterized by a nest of cells separated by a network of capillaries (the “zellballen” growth pattern). The tumor cells have abundant granular cytoplasm, which is more basophilic to amphophilic in pheochromocytoma and more eosinophilic in paraganglioma. Nuclear atypia may occur, but mitoses are usually low [8,83]. These tumors also exhibit some distinct immunohistochemical features. Since they are non-epithelial, they do not express keratins but express GATA-3. In addition, sustentacular cells surrounding the nest are S-100- and SOX10-positive [8].

### 7.1. Pheochromocytomas

Pheochromocytomas (intra-adrenal paragangliomas) may exhibit PAS-positive hyaline globules, lipofuscin inclusions, clear cells, oncocytic or spindle cells, in addition to the previously stated conventional histological characteristics [83]. These histological characteristics may pose diagnostic difficulties, although the immunohistochemistry pattern of positivity for neuroendocrine markers and GATA3 and negative for keratins strongly favors the diagnosis of neuroendocrine tumors rather than the epithelial origin. Additionally, the KI67 labeling index should also be assessed. Its value is usually low in pheochromocytoma (<10%), and higher rates could raise the suspicion of an alternative diagnosis [83].

All pheochromocytomas are considered malignant, with a potential metastasis range from 5% to 15%. Unfortunately, no histological features are formally used to predict the risk of metastasis even though multiple prognosis scores have been developed, such as GAPP (Grading System for Adrenal Pheochromocytoma and Paraganglioma), modified GAPP, PASS (Pheochromocytoma of the Adrenal Gland Scaled Score) [83] and COPPS (Composite Pheochromocytoma/Paraganglioma Prognostic Score) [104]. These scores evaluate histological features such as high cellularity and proliferation rates, comedo necrosis, and capsular or vascular invasion [105].

Another difficulty in managing pheochromocytomas comes from differentiating metastasis from multiple primary paragangliomas. Therefore, metastatic disease can be confirmed only when tumors occur in anatomic sites that cannot be affected by paragangliomas, such as lymph nodes and bones [6]. Even liver and lung involvement must be carefully assessed, since there have been a few isolated reports of primary paragangliomas in those regions [6,7]. Immunohistochemistry can aid the diagnosis as metastasis may lack S100 and SOX10 positive sustentacular cells [83].

### 7.2. Sympathetic Paragangliomas

Sympathetic paragangliomas affect the prevertebral and paravertebral sympathetic chains, sympathetic nerve plexuses, and sympathetic nerve fibers. They are most often located in the abdomen, retroperitoneum, pelvis, and thorax but can also affect the cervical sympathetic ganglia [83]. These tumors share very similar histological and immunohistochemical features with pheochromocytomas. Therefore, extensive imaging analysis should be used when one needs to differentiate a pheochromocytoma from an extra-adrenal paraganglioma due to a significantly higher risk of metastasis in the case of abdominal paragangliomas [105,106]. Additionally, paragangliomas with unusual locations, especially when they display less typical histological traits, might be confused for other malignancies such as renal cell carcinoma. Therefore, immunohistochemical analysis is mandatory to establish the correct diagnosis [107].

### 7.3. Parasympathetic Paragangliomas

Parasympathetic paragangliomas usually affect the vagus and glossopharyngeal nerve. Consequently, they are called “head and neck paragangliomas”. However, head and neck tumors can also include sympathetic paragangliomas arising from the cervical sympathetic chain or mixed paragangliomas [83,108].

Parasympathetic paragangliomas are most frequently located around the carotid body, followed by the middle ear and the nodose ganglion [83]. In addition, rare tumors can develop in the sella turcica, orbit, clivus, paranasal sinuses, mandible, parotid gland, nasopharynx, larynx, thyroid, and parathyroids [6,109]. Such rare cases may be misdiagnosed as primary epithelial NENs [83,110]. However, there are rare examples of epithelial NENs that do not express keratins, much as there are paragangliomas that focally express keratins. Nevertheless, immunohistochemical screening for keratins often yields an easy diagnosis [111]. Parasympathetic paragangliomas have a low potential for metastasis [110] but can be locally invasive and have a poor prognosis if surgical resection is not possible [112].

### 7.4. Composite Phaeochromocytoma and Paraganglioma

Composite phaeochromocytoma and paraganglioma display features of both phaeochromocytoma/paraganglioma and a neurogenic tumor such as ganglioneuroma, ganglioneuroblastoma, neuroblastoma, schwannoma, or malignant peripheral nerve sheath tumors (MPNST) [83,113,114]. To establish the diagnosis of composite phaeochromocytoma/paraganglioma, it is necessary to identify at least 5% of each tumor type. Thus, a few isolated neuron-like cells are not sufficient for a final diagnosis [83]. The most common subtype of the non-phaeochromocytoma/paraganglioma tumor is ganglioneuroma [79,115]. The prognosis of these tumors depends on the component with the highest grade. The risk of metastasis is significantly higher when the neurogenic component is represented by neuroblastoma or MPNST [83].

## 8. Conclusions

NENs are a diverse group of tumors that affect almost every organ; exhibit remarkable heterogeneity in terms of morphological traits, clinical presentation, and prognosis; and are widely regarded as malignant. In addition, these tumors are characterized by great heterogeneity in terms of origin, functional status, and aggressiveness. Consequently, in the NENs milieu, the diagnostician’s role in directing treatment has expanded and become more intricate. However, in the past decade, remarkable progress has been made in identifying common features for the entire spectrum of NENs due to advances in cellular and molecular pathology. As a result, it is now accepted that the majority of epithelial NENs can be divided into well-differentiated NETs and poorly differentiated NECs based on well-defined morphological and immunohistochemical criteria. Most NETs can be graded based on mitotic activity, while NECs do not need such a thing, as they always exhibit a high grade. Furthermore, NECs can be divided based on morphological criteria into SCNEC and LCNEC. Nevertheless, some sites, such as the pituitary gland, do not have well-established grading criteria. Still, the latest WHO edition of Endocrine and Neuroendocrine Tumors has considerably improved its classification for PitNETs. As far as non-epithelial NENs are concerned, they should be classified according to the affected site into pheochromocytomas, sympathetic paragangliomas, and parasympathetic paragangliomas, each subgroup with some particular features discussed in this chapter.

## Figures and Tables

**Figure 1 ijms-24-01418-f001:**
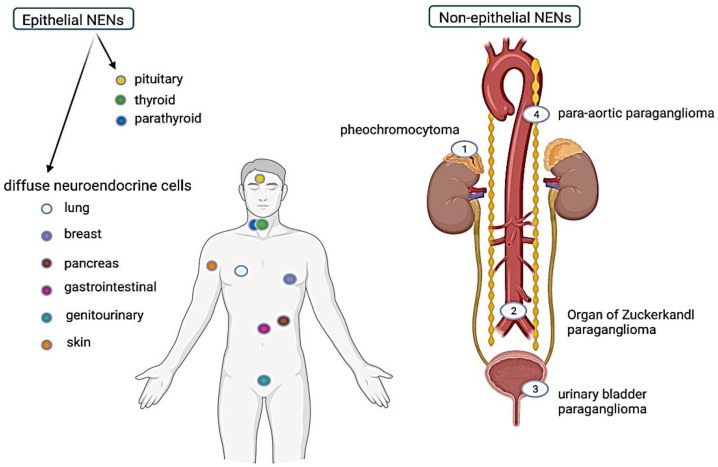
Overview of the epithelial and non-epithelial site of origin for NENs (Created with Bio.Render.com).

**Table 1 ijms-24-01418-t001:** Histopathological differences between NETs and NECs.

Histopathological Features	NETs	NECs
**Architecture**	“Organoid” proliferation with sporadic nests, trabecular patterns, and glandular or rosette development	Solid proliferation of less monomorphic cells with small-cells or large-cells
**Other features**	Usually, scant stroma with prominent blood vessels Occasionally, cell palisading at the periphery of the nests, amyloid stroma, calcifications with psammoma bodies	Areas of necrosis, apoptotic bodies
	NENs, especially NECs, can exhibit a non-neuroendocrine component with various differentiation, such as glandular or squamous	
**Cell features**	Usually epithelioid and closed-packed, rarely spindle and dis-cohesive cells Medium-sized cells with abundant cytoplasm and uniform round to oval nuclei with fine to coarsely granular chromatin and inconspicuous nucleoli (classic “salt and pepper” aspect)	Small-cell: scant cytoplasm, irregular, hyperchromatic nuclei with “salt and pepper” chromatin and severe nuclear molding Large-cell: abundant cytoplasm, vesicular nuclei with conspicuous nucleolus, possibly large and intensely eosinophilic
**Mitotic figures**	Absent or rare	Frequent

**Table 2 ijms-24-01418-t002:** The 2022 World Health Organization Classification of Epithelial Neuroendocrine Neoplasms.

Neuroendocrine Neoplasm	Classification	Defining Criteria
Lung and Thymus		
**Well-differentiated** **neuroendocrine tumor (NET)**	Typical carcinoid/NET G1	<2 mitoses/2 mm^2^ and no necrosis
	Atypical carcinoid/NET G2	2–10 mitoses/2 mm^2^ and/or necrosis (usually punctate)
	Carcinoid/NET with elevated mitotic counts and/or elevated Ki67 proliferation Index	Atypical carcinoid morphology and elevated miotic counts (>10 mitoses/2 mm^2^) and/or Ki67 > 30%
**Poorly differentiated** **neuroendocrine carcinoma (NEC)**	Small-cell lung carcinoma	Small-cell morphology and >10 mitoses/2 mm^2^
	Large-cell neuroendocrine lung carcinoma	Large-cell morphology, necrosis always present and >10 mitoses/2 mm^2^
**Gastrointestinal and** **pancreatobiliary tract**		
**Well-differentiated** **neuroendocrine tumor (NET)**	NET, grade 1	<2 mitoses/2 mm^2^ and/or Ki67 < 3%
	NET, grade 2	2–20 mitoses/2 mm^2^ and/or Ki67 3–20%
	NET, grade 3	>20 mitoses/2 mm^2^ and/or Ki67 > 20%
**Poorly differentiated** **neuroendocrine carcinoma (NEC)**	Small-cell neuroendocrine carcinoma	Small-cell morphology and >20 mitoses/2 mm^2^ and/or Ki67 > 20% (often > 70%)
	Large-cell neuroendocrine carcinoma	Large-cell morphology and >20 mitoses/2 mm^2^ and/or Ki67 > 20% (often > 70%)
**Upper aerodigestive tract and salivary glands**		
**Well-differentiated** **neuroendocrine tumor (NET)**	NET, grade 1	<2 mitoses/2 mm^2^, Ki67 < 20% and no necrosis
	NET, grade 2	2–10 mitoses/2 mm^2^, Ki67 < 20% and/or necrosis
	NET, grade 3	>10 mitoses/2 mm^2^ and/or Ki67 > 20%
**Poorly differentiated** **neuroendocrine carcinoma (NEC)**	Small-cell neuroendocrine carcinoma	Small-cell morphology and >10 mitoses/2 mm^2^ and/or Ki67 > 20% (often > 70%)
	Large-cell neuroendocrine carcinoma	Large-cell morphology and >20 mitoses/2 mm^2^ and/or Ki67 > 20% (often > 55%)

**Table 3 ijms-24-01418-t003:** Immunohistochemical markers used for diagnosing NENs.

	Chromogranin A, Synaptophysin	Ki67	Other Neuroendocrine Markers	Other Markers for Differential Diagnosis
**Lung NENs**	Mandatory for LCNECs May be negative in 15–20% of SCLCs	Desirable to be reported, >30% in carcinomas	CD56, INSM1, and/or POU2F3	CD45 for lymphoma or p40 for the basaloid version of squamous cell carcinoma
**Gastrointesntinal& ** **pancreaticobiliary NENs**	Chromogranin A: diffusely and intensely positive in NETs, but focally and weakly positive in NECs	Mandatory for grading NETs	SSTRs: diffusely positive in NETs but mostly absent in NECs	CK20: distinctive dot-like pattern for anal MCCs BCL10 and trypsin: for differentiating pancreatic NECs or G3 NETs from pancreatic acinar cell carcinoma
**Head&Neck NENs**	May be weakly positive in NECs	No clear cut-off value for NENs of the nasal cavity, paranasal sinuses, skull base, larynx, hypopharynx, trachea, and parapharyngeal space; Mandatory for grading MTC	INSM1: superior sensitivity for NECs; Calcitonin expression: a negative prognostic factor in MTC	For NECs: extensive analysis to exclude undifferentiated sinonasal carcinoma, NUT carcinoma, mucosal melanoma, embryonal rhabdomyosarcoma, neuroblastoma
**Breast NENs**	Positive but can also be expressed in other type of breast carcinomas	No clear cut-off value		The distinction between breast NENs and other entities should primarily be based on histological criteria
**Female Reproductive System NENs**	Positive expression of at least one: mandatory for diagnosis	No clear cut-off value	CD56: favors the diagnosis of NENs only in the presence of clear histopathological criteria	
**Urinary and Male Genital System NENs**	Usually expressed, but not mandatory for diagnosing SCNECs	No clear cut-off value	CD56: low specificity INSM1: superior sensitivity	
**Skin NENs**	Positive but can also be expressed in other skin carcinomas	No clear cut-off value		CK20 and/or NPF: characteristic dot-like pattern in Merkel Cell Carcinoma

**Table 4 ijms-24-01418-t004:** The 2022 WHO classification of PitNETs.

Tumor Type	Transcription Factor(s)	Hormone(s)	Keratin (CAM 5.2 or CK18)
**Somatotroph tumors**	PIT1	GH, α-subunit	Perinuclear
		GH	Fibrous bodies (>70%)
**Lactotroph tumors**	PIT1, ERα	PRL (paranuclear)	Weak or negative
		PRL (diffuse cytoplasmic)	Weak or negative
**Mammosomatotroph tumor**	PIT1, ERα	GH (often predominant) PRL, α-subunit	Perinuclear
**Thyrotroph tumor**	PIT1, GATA2/3	α-subunit, βTSH	Weak or negative
**Mature plurihormonal PIT1-lineage tumor**	PIT1, ERα, GATA2/3	GH (often predominant), PRL, α-subunit, βTSH	Perinuclear
**Acidophil stem cell tumor**	PIT1, ERα	PRL (predominant), GH (focal/variable)	Scattered fibrous bodies
**Immature PIT1-lineage tumor**	PIT1, ERα GATA2/3	GH, PRL, α-subunit, βTSH	Focal/ Variable
**Corticotroph tumors**	TPIT (TBX19), NeuroD1/β2	ACTH and other POMC derivatives	Strong
			Variable
			Intense ring-like perinuclear
**Gonadotroph tumor**	SF1, ERα GATA2/3	α-subunit, βFSH, βLH	Variable
**Unclassified plurihormonal tumors**	Multiple combinations	Multiple combinations	Variable
**Null cell tumor**	None	None	Variable

## Data Availability

Not applicable.

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
