# Peer review of "The Complex Histopathological and Immunohistochemical Spectrum of Neuroendocrine Tumors—An Overview of the Latest Classifications"

_ijms, 2023, doi:10.3390/ijms24021418_

Round 1
Reviewer 1 Report
this is a very clear and complete review in respect to recent changes in the classification of neuroendocrine tumors. I would like to make to considerations:
1) Neuroendocrine tumors are considered malignant. Pituitary adenomas has been regarded as benign tumors, with metastasis occurring rather rarely, although some tumors may present aggressive behavior. How this reclassification of pituitary tumors as PitNETs change the way these tumors are viewed? They should be considered as malignant as well?
2) According to the latest WHO classification, pheochromocytomas are classified as intra-adrenal paraganglioma. In the manuscript it seems that it is an entity different from paragangliomas.
Author Response
We want to thank the reviewers for their observations. The revised paper contains changes according to comments. As indicated, we highlighted what was added to make it easier to track our changes. In addition, the text was subjected again to grammar and spelling checking -all modifications are highlighted as indicated.
Response to Reviewer 1 comments
Neuroendocrine tumors are considered malignant. Pituitary adenomas has been regarded as benign tumors, with metastasis occurring rather rarely, although some tumors may present aggressive behavior. How this reclassification of pituitary tumors as PitNETs change the way these tumors are viewed? They should be considered as malignant as well?
The name "adenoma" has been changed to "pituitary neuroendocrine tumor" (PitNET), which is a significant terminology change from the previous version of the WHO classification. Since the adenohypophysis's hormone-secreting cells are neuroendocrine cells, their tumors are classified as neuroendocrine neoplasms. In addition, based on the infrequent occurrence of metastatic behavior, they have been categorized as adenomas for a long time. Adenomas are benign lesions that do not usually represent a medical issue in terms of treatment and prognosis. In reality, pituitary tumors are often invasive neoplasms that, like carcinomas, may infiltrate nearby tissues. Furthermore, neither morphologic nor molecular characteristics can predict if and when they spread metastatically. Therefore, according to conventional nomenclature, the first diagnosis is "adenoma." The diagnostic is modified to "carcinoma" only after the metastasis has been found. Finally, pituitary tumors that cannot be removed surgically are treated with the same type of therapies as neuroendocrine tumors found in other body parts. This is another significant aspect of the management strategy. For these reasons and according to the 2022 WHO Classification of Pituitary Tumors (10.1007/s12022-022-09703-7) the last nomenclature is more beneficial to better stratify patients with tumors arising in the pituitary gland.
According to the latest WHO classification, pheochromocytomas are classified as intra-adrenal paraganglioma. In the manuscript it seems that it is an entity different from paragangliomas.
We want to thank the reviewer for this remark. Pheochromocytoma and paraganglioma are tumors that come from the same type of tissue. Pheochromocytoma is a tumor of the adrenal medulla (the center of the adrenal gland), while paragangliomas form outside the adrenal gland. To avoid misunderstanding, and according to the latest nomenclature proposed by WHO, we have added in the non-epithelial neuroendocrine tumor section the terminology of intra-adrenal paraganglioma.
Reviewer 2 Report
The Histopathological classification of neuroendocrine neoplasms is quite challenging even for clinicians specializing in oncology so an overview of the NENs latest Classifications would be very useful.
The article tackles most of the important issues concerning latest NENs classifications, but unfortunately in parts it is written in a way that is difficult to follow.
The suggestion for authors:
Ø All chapters should be re-written in more comprehensive way
Ø Could you provide the information of the types of NEN (described in introduction) in table or chart?
Ø For chapter 2 please consider providing (in table) information about general histopathological differences between NETs and NECs
Ø Chapter 4
- Line 195: “For example, patients with NECs present with nonspecific symptoms of an aggressive systemic illness and have undetectable levels of chromogranin A, while patients with NETs often present with carcinoid syndrome induced by biogenic amines hypersecretion [11, 36].” Chapter 4 refers to the differentiation of NET G3 and NEC – it sounds clinically doubtful, could you clarify what percentage of patients with NET G3 have carcinoid syndrome?
- Line 198 “In imaging studies, NETs can be detected on functional imaging, although they could remain undetected on PET/MRI. Contrarily, functional imaging is of little use for identifying NECs, which can be detected on PET/MRI [38].” - according to my knowledge both PET modalities (PET/CT and PET/MRI) are a used for functional imaging and are quite sensitive in NETs with good SSTR expression -please clarify
- Please consider for this chapter discussion about the differences in patients management in case of NETsG3 and NECs as well as differences in their prognosis
Ø Could you consider providing in chapter 6 (of specific types of NENs) information about recommended IHC (in table)?
Author Response
We want to thank the reviewers for their observations. The revised paper contains changes according to comments. As indicated, we highlighted what was added to make it easier to track our changes. In addition, the text was subjected again to grammar and spelling checking -all modifications are highlighted as indicated.
Response to Reviewer 2 comments
All chapters should be re-written in more comprehensive way
The information has been incorporated in two additional tables in the revised manuscript to make the content more comprehensible. In addition to this, more data has been provided in the text for several chapters. Furthermore, a figure was inserted in the introduction chapter to improve the present article.
Could you provide the information of the types of NEN (described in introduction) in table or chart?
To facilitate the comprehension of the fundamental histological classification of neuroendocrine tumors and their various sites of origin, a figure has been included in Section 1.
For chapter 2 please consider providing (in table) information about general histopathological differences between NETs and NECs
As recommended, we added in chapter 2 a table that represents an overview of the histopathological differences found between NETs and NECs.
Line 195: “For example, patients with NECs present with nonspecific symptoms of an aggressive systemic illness and have undetectable levels of chromogranin A, while patients with NETs often present with carcinoid syndrome induced by biogenic amines hypersecretion [11, 36].” Chapter 4 refers to the differentiation of NET G3 and NEC – it sounds clinically doubtful, could you clarify what percentage of patients with NET G3 have carcinoid syndrome?
We want to thank the reviewer for this remark. In the chapter dedicated to the comparative analysis of the two histopathological entities (NET G3 vs. NEC), we have added data regarding the functional status and the dynamic of the biomarkers. To our knowledge, no studies have evaluated the presence of carcinoid syndrome in the NET G3 category versus NEC. Furthermore, clinical results, not pathologic features or immunohistochemical profiles, determine the tumor's functional state.
Line 198 “In imaging studies, NETs can be detected on functional imaging, although they could remain undetected on PET/MRI. Contrarily, functional imaging is of little use for identifying NECs, which can be detected on PET/MRI [38].” - according to my knowledge both PET modalities (PET/CT and PET/MRI) are a used for functional imaging and are quite sensitive in NETs with good SSTR expression -please clarify.
We thank the reviewer for this observation. Modifications have been made to the diagnostic imaging section to clarify the content. In this sense, we highlighted that FDG-PET is recommended over somatostatin receptor imaging for NEC.
Please consider for this chapter discussion about the differences in patients management in case of NETsG3 and NECs as well as differences in their prognosis.
To separate the two histopathological entities, data on survival and response to treatment were included in section 4. We did not believe it essential to elaborate on the clinical characteristics of NET (either NET G3 or NEC) since this issue has been widely covered in the literature, and also because our paper provides a summary of the most recent classifications with emphasis on the complex histopathological and immunohistochemical spectrum of neuroendocrine tumors.
Could you consider providing in chapter 6 (of specific types of NENs) information about recommended IHC (in table)?
As suggested, we have added at the end of section 6 a comprehensive table (table 4) that includes the immunohistochemical markers used for diagnosing NENs of various locations. Thus, table 3 recaps the most frequently used immunohistochemical markers in practice both for NETs of the lung and gastrointestinal tract as well as for rare locations such as the reproductive system.
Round 2
Reviewer 2 Report
The authors have sufficiently improved the manuscript. I have no further comments.